# Managing Tourist Destinations According to the Principles of the Social Economy: The Case of the *Les Oiseaux de Passage* Cooperative Platform

**Blanca Miedes-Ugarte** [1] **, David Flores-Ruiz** [1,*] **and Prosper Wanner** [2]

[1]  Departamento de Economía, Universidad de Huelva, 21071 Huelva, Spain; miedes@uhu.es
[2]  Centre d'études en Sciences Sociales CESSMA, Université Paris Diderot,-Inalco,-IRD, 75205 Paris, France; pwanner@lesoiseauxdepassage.coop
*   Correspondence: david.flores@dege.uhu.es

**Abstract:** Two key factors that need to be considered in the management of tourist destinations are the model of governance that is adopted and the kind of technology that is employed. Poor decisions in this regard can have serious consequences for sustainability in accordance with the 2030 Sustainable Development Goals (SDGs). This case-study analyses the outcomes of an axiological and practical application of cooperative principles, with appropriate technological support, to the territorial governance of travel and hospitality services. It focuses on the implementation of an R&D+i project to create an online cooperative platform managing 40 destinations. The practical application of these principles is seen to require a shift in perspective, not only in terms of the conception of territory, going from a space of purely capital valorisation to a commonly-held co-constructed heritage asset, but also in the approach to the use of technology, which favours peer-based collective intelligence over blind artificial intelligence. The most notable features of the model identified by the findings are increased proximity and inclusiveness on the part of users, and enhanced sustainability. With respect to the technological platform, the analysis indicates that it is scalable and replicable, as demonstrated by the growth from 7 to 40 destinations in a single year.

**Keywords:** tourism destination governance; social economy; sustainable development; collaborative platforms; hospitality territories

## 1. Introduction

At the time of writing, tourism is facing an unprecedented worldwide crisis as a result of the COVID-19 pandemic. According to a report published by the World Tourism Organization (WTO) in May 2020, international tourism is expected to decrease by between 60% and 80% in 2020, with a reduction of 22% in the first quarter [1]—all this in the midst of a severe worldwide recession, in which the recovery of international tourism is expected to be slow.

Against this backdrop, there is an urgent need to reflect on the future of a sector in which since the 1970s, according to [2,3], there has been considerable debate not only around the negative impact of tourism on the environment and the limitations of its capacity to generate economically and socially inclusive developments in host territories, but also around the issue of global elitism (according to the WTO report cited above, only 18% of the world's population have travelled as tourists, an observation which reflects both widespread income inequality and the restrictions placed upon international travel by visa requirements).

The current tour operator model, based on packaging transport (*ever further, ever faster*) together with standardized accommodation (in constructions producing a high degree of pollution and

demanding intensive consumption of air-conditioning and water), is unsustainable. It targets solely consumer economies and establishes discriminatory prices according to profiling algorithms. In short, as the current health crisis has highlighted, it is a self-serving industry which fails to recognize other kinds of journeys and stays, and makes no consideration for a range of people who might need these services, such as the migrant workforce, refugees, students, and those with long-term illnesses [4]. This situation is especially worrying in view of the current tendencies, which according to the figures in [5], forecast 250 million climate refugees by 2050.

From an environmental perspective, [6] estimates that, in terms of transport and accommodation, tourism accounts for around 8% of global warming. Further, nearly three-quarters of the $CO_2$ emissions relating to tourism derive from air traffic and road transport (40% and 32%, respectively), followed by accommodation (over 20%) and cruises [7].

Essentially, in terms of the short and long-term structural ramifications, the current crisis provides a window for tourism to reflect on how it can move on from the crisis. Lessons will need to be learned and, above all, models of tourism will need to be developed which follow the UN Agenda 2030 by putting well-being, inclusive prosperity and sustainability at the centre of their concerns—models in which local issues and cooperative forms of management typical of the social economy provide responses to the current problems and global trends.

This much-needed overhaul of the tourism system can be best achieved through the implementation of evidence-based multidisciplinary research in conjunction with innovative practices. There is a need, too, for a broader imaginative outlook which finds inspiration in the positive image of a better future, and which, looking beyond the immediate focus on problems and necessities, sees new opportunities in existing resources.

For such ideas to be put into action, it is useful to identify "seeds"—small-scale projects already in existence, which suggest what the "positive future" might look like, and which, if circumstances permit, could become the norm [8]. These seeds adopt new forms of thinking, doing and being at the margins of the mainstream, and have achieved interesting results in terms of their social, economic, cultural and environmental impact. They suggest potential approaches somewhat underexplored by current models of tourism, but which if taken up and imitated, could result in a substantive difference in terms of sustainability and social occasion [9,10]. In this respect, such seeds can be regarded as challenging the notion of tourism itself. They place emphasis on the experiential nature of travel, on the creation of fairer relations among all of the actors in the tourism dynamic, and on more meaningful and human interchanges between visitors and local inhabitants.

This study aims to contribute to the debate on future directions in tourism, especially necessary in the context of the COVID-19 pandemic and its consequences, by reporting on a case study based on one of these "seeds", namely the cooperative online platform "Les Oiseaux de Passage".

The theoretical framework for the analysis of the case in question is structured in the following way. (a) The paper first discusses the transformations that the tourist sector has undergone in recent years in response to the worldwide increase in tourism based on sharing platforms—among which Airbnb is a much-imitated model—and the resultant impact on tourist destinations. The authors argue that these shifts have precipitated an urgent need for more sustainable and inclusive models of development, which are regulated and managed from the supply approach rather than demand. (b) This is followed by a discussion of what are commonly termed "smart destinations" and how these are managed, as an example of a mainstream response to these challenges. The second half of the paper focuses on the case under analysis. The Figure 1 presents the reseach overview.

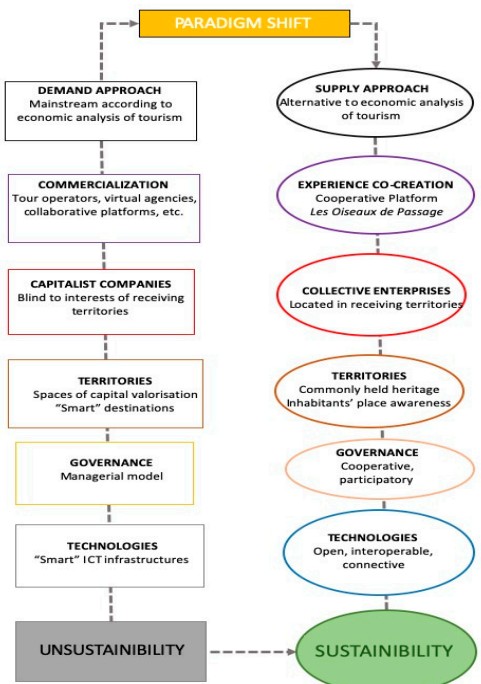

**Figure 1.** Paradigm shift in managing tourist destinations. A research overview; Source: elaborated by authors.

## 2. An Unsustainable Model: Sharing Platforms and the Management of Tourist Destinations.

In recent decades, the development of Information and Communication Technologies (ICT) has restructured the value chain within the sector. Since the 1950s, as a result of the development of Global Distribution Systems (GDS), large scale tour operators have dominated the industry by controlling the greater part of the value chain (accommodation, transport, travel agents and so on). The sector expanded with the advent of information technology, in terms of both financial management and the creation of client profiles, and the standardisation of products accordingly [11]. Indeed, these elements can be considered the precursors of many practices which were subsequently developed on the internet in the 1990s. Since then, with the revolution of the worldwide web and virtual platforms, the power of the tour operators has declined to the benefit of other players, such as online travel agents, booking centres, sharing economy platforms, and, increasingly, tourists themselves via the appearance of new concepts of transaction such as "co-creation" and "prosumption" (production by consumers).

In this context, according to [12], the value of an experience is increasingly created and co-created during the process of planning and purchase, as well as during the enjoyment and recollection of the trip itself. Indeed, tourism is considered the example par excellence of the process of co-creation, testimony of which can be seen in the fact that in 2014, for example, more than half of tourists organised their trip online on the basis of recommendations from friends or relations [13].

Traditionally, local supply is determined by global demand, to which it responds in reactive-adaptive fashion. To a certain extent, even global demand itself is reactive, since, as [14] observes, the major capitalists and techno-structures have a limited influence. They are not creative: "Their action is limited to making the most of innovations and fashions" [14].

Despite this, it can be said that tourism has traditionally been configured largely from the top down, in response to a demand created in part autonomously, but nevertheless under the control of the large tour operators, and that this determines the local dynamics of popular destinations. That is to say, the function of producing tourism tends to be outside the control of the destinations themselves; in the final analysis, the tourism product is created by the tourists, or else by foreign tour operators, guided by their own interests, not coincident with those of the host area. In fact, over a third of the

income/profits generated by the sector wind up in those countries controlling tourist destinations at a global level [15,16].

As shall be seen below, all of these problems have been exacerbated by an increase in online sharing platforms, which have only served to consolidate the focus on demand, creating additional tensions in the traditional model in host territories.

### 2.1. The Discourse of the Sharing Economy: The Truth Behind Airbnb's Rhetoric

Innovations in ICT (internet, mobile networks, social media, apps and web platforms, etc.) have seen a blurring of the distinction between producers and consumers within certain sectors, giving rise to the concept of "prosumer", someone who is simultaneously a producer and consumer of a product [17,18]. Tourist production, formerly almost exclusively the domain of tour operators, is now a role in which the tourist can participate via online platforms.

In fact, it was [19] which popularised the use of the concepts of collaborative consumption and the shared economy in reference to a series of activities and practices involving the exchange of goods and services between private individuals via online platforms (P2P).

In this regard, the business that has achieved the greatest success and has become the model to follow is that of Airbnb, which in scarcely a decade has grown into the world's largest provider of accommodation rentals, spanning 191 countries and taking close to 400 million bookings since 2008, according to a study by [20], overtaking all of the largest hotel chains on the way.

Yet, as multiple studies have shown, Airbnb's success has not come without generating an increasing degree of ill-feeling among residents of many destinations, who complain that the socio-economic and environmental impacts are a far cry from the positivist discourse of studies carried out under the auspices of the key players in the sector themselves, such as [21], and even the European Commission [22].

According to [23], much of the discourse pointing to the benefits and positive impacts of such platforms is founded on aspects which have been called into question by various studies, among which the following can be highlighted:

- In contrast to claims of disintermediation as a result of the collaborative practices of the digital economy, according to [24], not only are intermediaries not eliminated, but they take on unprecedented importance and power, which, due to the effects of the internet, display clear monopolistic tendencies.

- The platforms keep very strict control over the information that users can exchange in order to prevent any communication and ultimate economic transaction being conducted through alternative channels, going so far as to charge a commission of up to 20% of the published fee. In the view of [25], such platforms exercise an authoritarian control over what actions users can do on them.

- These players take on the role of producers of tourism, oriented towards demand, a role historically occupied by the large tour operators. In addition, they allow tourists themselves to participate in the configuration of the tourist products on offer, giving users the possibility to co-create them (in terms of organising and planning the outward and return journeys, for example). In this respect, they are genuine capitalist enterprises engaged in the production or facilitation of tourism products (experiences), and shaping them according to their own interests, in much the same way as the large tour operators have always done.

- With respect to the claim that such platforms have the ability to promote a welcome redistribution of tourist destinations across different locations [21] (Airbnb, 2014), numerous studies have found that they have instead exacerbated the flow to traditional tourist hotspots [26–29].

- Another much repeated claim of the collaborative consumption discourse is that it results in an even distribution of extra income for the hosts. Again, research suggests that this is true only for such individuals as are able to offer accommodation in properties where they are not

resident—ranging from buy-to-let flats to second homes—and for whom the rental either wholly or partially constitutes their living. For example, according to [29], only 10.54% of rentals in Mallorca involve domiciles, that is with the owner and guest sharing space, with the remaining 89.46% involving rentals of the complete property. The conclusion to be drawn is that the vast majority of private rentals are by those with sufficient resources to rent out a property in which they are not resident for anything between several months to the whole year.

- Related to the above, another frequent complaint about collaborative platforms concerns the part they play in the so-called gentrification of over-subscribed areas. The effect is two-fold: the rental market reduces the housing stock available for residential purposes, while at the same time driving up the rents on what residential properties there are. The result, as described in sources [29] and [30], is that locals can no longer afford to live in their own town.

- A commonly occurring trope of the discourse is that of providing a quality experience, a new way of doing tourism radically different from traditional modes, which brings added value. Nevertheless, there is no getting away from the fact that the business model is low-cost, exploiting the possibilities offered by technology to enable the consumer to put together their own product and so reduce costs, but inevitably introducing job insecurity and impoverishment into the capitalist society, alongside high environmental costs [20].

- The trope of the quality alternative experience, in which, through the inclusivity of P2P, the tourist is transformed into a guest among the local residents, is also given the lie by the simple fact that, as reported by [20,29], in the majority of cases, the host is neither local nor resident, and the rental approximates a hotel more than a private residence, with the result that the tourist rarely fulfils the promise of participating in the local lifestyle.

- Finally, according to [23] (2019), platforms such as Airbnb have been lobbying European institutions over the last few years to make changes to the legislative framework of short-term lets that benefit their business model. These aspirations were articulated in the report "A European Agenda for the Collaborative Economy" [22], one of the key objectives of which was legitimise collaborative discourse in a document of political and symbolic significance.

In short, where all the above studies concur is in the unsustainability of this model of development for tourist destinations. The current situation is no more than the natural outcome of the capitalist model of the tour operators applied to the digital economy—or, more precisely, of the tourism-producing function these players have always applied.

Hence, as long as this productive function remains in the hands of the major capitalist companies, out of the reach of the actual tourist destinations, the impact on these territories is likely to be more negative than positive, from both the socio-economic and environmental perspectives. Not surprisingly, this model of tourist development has only increased the negative impacts on the destination wherever it has been implanted, as the extensive literature on the subject testifies.

### 2.2. The Need for a Destination-Driven Approach Focused on Supply. The Debate Around "Smart Destinations" and Their Governance

It is at this point, when it comes to presenting investment strategies for the development of sustainable tourist destinations, that the argument put forward by [31] takes on a special importance. It proposes that, while the analytical model based on demand leads those involved to produce "for tourism" (hotels, restaurants, infrastructure, leisure activities, etc., whether for tour operators or for large collaborative platforms), its counterpoint based on supply leads them to produce "tourism" (creation of the most complete and fulfilling sojourn possible in the actual destination). This strategy of attempting to develop the function of production of tourism in the area itself enables the destination to control for itself, to a greatly increased extent, its own model of tourist development and the impact this might have on the area.

The relationships between territory and tourism are multiple, complex and varied, and over the years the literature has tended to treat them from a territorial perspective. According to [32], this is due to:

- the great variety of sectors and activities involved, which are located within the same territory and maintain important complementary interrelationships;
- the need to take into consideration both public and private stakeholders within an individual area, all of whom need to establish a cooperative matrix for the hospitality experience to work;
- the fact that, despite the involvement of all levels of public administration—state, regional, district and local—in the development of destinations, the ideal level for implementing tourism policies is that of district and/or local, as this is where the tourism actually takes place. This can be seen, for example, in the shifting emphasis of tourism policies in Spain, which according to [33], have seen increasing importance over time given to the local level;
- the need to provide visitors with a series of public and quasi-public goods and services—infrastructure, natural resources, sociocultural heritage, security, cleaning, etc.—which again are best managed collectively by stakeholders in situ (management of common goods and services);
- the phenomenon by which, as [34] notes, a territory not only represents the locus of tourist activity, but also an argument for the same, hence creating the need to balance its development against its conservation.

One concept which recognises the complexity of the relationship between tourism and territory, and which mediates this relationship through the use of ICT, is that of the Smart Tourism Destination (STD). This concept, according to [35], can be defined as "an innovative tourist destination, built on an infrastructure of state-of-the-art technology guaranteeing the sustainable development of tourist areas, accessible to everyone, which facilitates the visitor's interaction with and integration into his or her surroundings, increases the quality of the experience at the destination, and improves residents' quality of life". While this definition succinctly captures the aims of "smart tourism", the authors of [36] provide a clearer indication of the means by which this might be achieved. For them, a smart tourist destination is "an innovative space founded on an area and a state-of-the-art technological infrastructure. This area is committed the local environmental, cultural and socio-economic factors, and is equipped with an intelligent system capable of gathering and analysing information procedurally so as to understand events in real time, with the aim of facilitating both the interaction of visitors with the environment, and the decision-making procedures of local managers, thus increasing efficiency and substantively improving the quality of the tourist experience".

This latter definition brings us closer to the heart of STDs and includes all the key elements which mark them out as innovative and which, as indicated in [37], constitute the concept of Territorial Intelligence (TI). Specifically, these elements are technology, sustainability (environmental, cultural and socio-economic), information processing, and knowledge and efficiency generation.

However, the use of the concept of intelligence in the Spanish context, as [38,39] conclude in their study into the application of the principles of STD in Spain, tends to be limited almost exclusively to the implementation at the destination of advanced technologies, whilst the other facets receive far less attention.

In this respect, rather than being genuinely smart, such destinations employ technological intelligence, but do not develop true territorial intelligence [37,40]. In effect, the technology is not implemented within a framework of participative and collaborative construction of territorial knowledge, and the aim is not to enhance the management of resources by local residents and the main stakeholders according to the four cardinal principles of sustainability, multidimensionality, partnership and participation, as identified by [41].

To promote the development of territorial intelligence, it is not enough to merely upgrade the ICT infrastructure with smart systems and to contract the services of specialists in knowledge management.

What it does require, as [42] points out, is a shift in relational dynamics in order to bring about changes in the structures, processes and collective rules involved.

All of which underlines the decision-making process in the area in question—the territorial governance—and the need for this to be as participative as possible so that, as [43,44] argue, the processes of collective intelligence can be channelled. However, studies such as [38,45] illustrate that the facet of governance, such a key component of territorial intelligence, is largely overlooked during the process of converting traditionally oriented destinations into smart destinations. Unsurprisingly, as [46] note, the majority of tourist locations claiming to have processes of governance in place, in reality fulfil very few of the principles that characterise it.

The implications of such observations are that many tourist destinations initiate their development without taking into account in any tangible and decisive way the involvement and integration of the full range of stakeholders—businesses, civil society, tourists, research centres and public administration. This deficit, as shall be seen below, is fully addressed by the case under consideration, from which, as we see it, various conclusions and lessons can be drawn, above all if its chief defining features are measured against the theoretical framework outlined above.

## 3. Methodology

The methodology followed in this study is case-study analysis, namely the cooperative online platform "Les Oiseaux de Passage".

One of the aspects which makes this case of particular interest is its dimension of R&D+i, as it concerns the second Cooperative Society of Public Interest to be recognised in France as a "Jeune entreprise innovante (JEI)" (Young Innovative Company) for its social innovation [46]. The recognition comes with a 7-year applied research grant to fund collaboration among the four applicant research centres, and a subsidy of 75% of the salary of a doctoral student for three years, whose research college controls 30% of the vote in the cooperative.

This paper presents an analysis of the defining features of this recent, radically innovative action-research initiative based on co-creation and co-management according to the cooperative principles of the social economy, within the ambit of territorial governance of tourism services.

The starting hypothesis considers that the case under analysis is a good illustration of a paradigm shift in the conceptualisation of tourism and how it can be locally managed at the destination sites following the cooperative principles of the social economy. Drawing on this analysis, a series of useful guidelines for developing more socially inclusive and environmentally sustainable models of tourism is presented.

In exploratory studies such as this, an appropriate methodological approach is that of the case-study [47–52], which has been adopted here. After the theoretical foundations have been established, documentary sources regarding the context in which the cooperative was set up, its subsequent development, and the essential features of the online platform are analysed. The study also draws on the direct experience of one of the founders of the cooperative, who is also a member of the research-action team and co-author of this paper, for first-hand description and analysis.

The following section describes the online platform on which the case is based, the principles of the social economy around which it is structured and the essential characteristics of its cooperative governance [53]. This is followed by a discussion of those features which, in the view of the authors, have the greatest potential to transform the sector, specifically (a) the conceptualization of territory as a collectively held asset in the development of authentic territorial intelligence [37,53]; (b) the principle of enhancement of human relations in the choice of technology [54]. The final part of the paper sets out the most salient conclusions to be drawn.

## 4. The "Les Oiseaux de Passage" Platform: Cooperative Governance in Tourist Locations

### 4.1. The "Les Oiseaux de Passage" Platform

The collaborative online platform "Les Oiseaux de Passage" [55] was set up in France at the beginning of 2019 with the aim of encouraging the travel and hospitality industry to embrace sustainable development in the communities and destinations involved.

The collaborative platform was originally given an impulse by another cooperative, the Hôtel du Nord residents' cooperative, set up in 2011 in the northern districts of Marseilles in response to three economic imperatives: (a) to find an economy capable of keeping alive the history of the area in a sustainable rather than ad hoc manner; (b) to create an economy for those engaged in supporting these elements of heritage, especially the most economically vulnerable (women and children); (c) to promote contact with others through the history, the people and the locations across these districts of the city. The social ambitions of the cooperative were to bring economic benefits to the heritage of the 15th and 16th districts of Marseilles, to ensure that it remained a living heritage, and to improve the lives of those who lived and worked there. The Hôtel du Nord cooperative comprises a network of 60 accommodation units (guest rooms, independent flats and cottages), a community of hospitable hosts, keen to act as guides to their home terrain and to share its stories and traditions, and a hundred or so heritage trails around the area radiating out from a starting point at the heart of the northern district.

In January 2016, three cooperatives—Hôtel du Nord, Ekitour and Point Carré—along with the Minga network and five individual members founded "Les Oiseaux de Passage", a Cooperative Society of Public Interest (from the French 'société cooperative d'interet publique' or SCIC). The purpose of this SCIC was to research and develop a set of online tools for promoting and commercializing an alternative kind of hospitality, which went beyond solely accommodation to include workshops and activities that enabled the hosts to share their expertise and passion for their heritage. In doing so, the platform was intended to contribute to the economic, social and cultural development of the participating areas, improving the quality of life of those living and working there, enriching people's awareness of other regions and cultures, and, more generally, reinforcing the principles of democracy, rule of law and human rights.

In total, some 500,000 euros were invested between the research and development stage (2017-2018) and its launch in 2019. The financing was split evenly between private investment, bank loans and public funds targeted at innovation (Research Tax Credit [56] on investment and activity in 2019 and a CIFRE [57] grant for a three years PhD research project, 2019-2021).

The R&D+i agencies involved in the development of the platform represented different research teams: Laboratoire CESSMA Anthropologie (Centre d'études en sciences sociales sur les mondes africains, américains et asiatiques). Economie de l'altérité Université París Diderot; Programme TAPAS (There Are Platforms as AlternativeS) programme, Centre d'économie et de gestion, Université Paris 13; Laboratoire CRIEF, Économie Social, Tourisme Social, in conjunction with the Faro Convention Action-Research Network of the Council of Europe [58].

The "Les Oiseaux de Passage" platform today brings together 220 individuals and organisations as hosts from across 52 communities (from the starting nine), offering 85 accommodation options, 220 activities, 536 *bon plans* and 70 'creations' (local produce, guidebooks and the like) spread across 40 destinations (form the original seven) [55]. At the current time, the platform is expanding to other locations in France and beyond, including Spain, Italy, Belgium, and even Algeria. Each community endeavours to offer a series of hospitality services and experiences rooted in the traditions and culture of that location.

The initiatives on offer usually derive from a group of people or hosts, connected around an area through a common narrative. Each publishes their offer with a description on the website, and links it to others to form a chain of itineraries, cultural destinations and shared stories. With a detailed knowledge of the area, each community can offer an expert, full and varied experience.

The platform incorporates a collaboration tool enabling contributors to put together a menu of cultural destinations, routes and stories to inspire the curious visitors to come and try them for themselves. For their part, travellers can organise their trip without the need to visit multiple websites, deciding on their own experiences from the range of activities on offer. In this regard, the platform is configured as a true factory of stories for communities and a factory of journeys for travellers, whether alone or in groups.

*4.2. Characteristics of Cooperative Governance Applied to the "Les Oiseaux de Passage" Platform*

As suggested above, the governance of such tourism destinations is one of the more complex aspects to manage, with very few successful cases reported in the literature [46].

Despite this pessimistic panorama, the application of the principles of coordination, collaboration and cooperation among stakeholders along the tourism value chain, as [59] point out, can facilitate consensus and learning during the initial phases of planning and setting up the tourism destination, thus underlining the importance of these processes later in the development of the sustainable tourism project.

Further, as [60] observe, there is a clear correlation between the dynamics of the relationships among the different stakeholders involved in managing the destinations and how this common cause develops over time: the more productive the relationships, the more successful the project.

In its policy document on Governance for Sustainable Human Development, the United Nations Development Program (UNDP) establishes a series of key principles of good governance: participation, rule of law, transparency, responsiveness, consensus orientation, equity, effectiveness and efficiency, accountability and strategic vision. It is not enough that some of these principles should be enforced, rather all of them should be applied without any entering into conflict with each other [61,62]. In order to implement processes of sustainable development in tourism, it is therefore necessary to apply a transparent and self-evidently fair governance which reflects community and social economy management methods based on participation, self-management of much of the development process, democratic decision-making, and the equitable distribution of resources and benefits, according to [63,64].

These principles coincide with those contained in the Social Economy Charter of the European Standing Conference of Co-operatives, Mutual Societies, Associations and Foundations, namely: primacy of people and of the social objective over capital, voluntary and open membership, democratic control by the membership, the combination of the interest of members/users and society (general interest), the defense and application of the principles of solidarity and responsibility, autonomous management and independence from public authorities, reinvestment of the essential surplus to carry out sustainable development objectives, services of interest to members or of general interest [65].

In keeping with these cooperative principles, and in contrast to the capitalist model, by which collaborative platforms belong to large groups of shareholders, the "Les Oiseaux de Passge" platform devolves decision-making to all of the major stakeholders. The hospitality communities and producer networks represent a statutory majority (with 50% of the voting rights). The other members of the cooperative comprise disseminators of the services, such as travel agencies, consumer cooperatives, travellers' associations, and so on (with 20% of the votes), and the universities and researchers (two researchers and five contact personnel, with 30% of the votes). These latter two groups have representation (directly or through their networks), they can freely request to be part of the cooperative, and can participate democratically in its governance. By the same token, and again unlike the corporate platforms, "Les Oiseaux de Passge" actively sets out to reappropriate, on behalf of the workers and users, the latest developments in ICT applied to the hospitality and travel sector.

Table 1, below, lists the different cooperative principles and describes how these are specified, applied and put into practice by the management system of the platform.

**Table 1.** Cooperative principles of the Les Oiseaux de Passage platform.

| Cooperative Principles | Realization on the Les Oiseaux de Passage Platform |
|---|---|
| **Primacy of the social objective over capital** | The technology is at the service of people (guests and hosts). |
| **Transparency and equity** | Transparent, fair and economical fees: each professional selects the pricing scheme most appropriate to their circumstances, and benefits directly from the different services included. Payment is by annual fee irrespective of volume of sales. (By contrast, the major platforms charge up to 20% of sales.) |
| **Voluntary and open membership** | Hosts, grouped into communities, can request voluntary and open membership of the cooperative. Other members of the cooperative are distributors of the offer (travel agents, workers' committees, and tourist associations), universities and researchers. |
| **Democratic control by the membership** | The communities are a statutory majority, represented directly or through their networks, and participate democratically in its governance. |
| **The combination of individual and general interests** | Each community and each of its members has their own website where they can freely promote themselves and share contacts with other communities and destinations. This constitutes a collective response to the diversity and evolution in travel patterns and web use. |
| **Cooperation** | The platform provides a tool for collaboration and, particularly, cooperation between hosts in different communities and destinations. |
| **Autonomous management and independence from public authorities** | The cooperative has its own structure, objectives and management criteria, independent of the public administration, and managed by its members. |
| **Collective interest of the cooperative** | This kind of platform cooperativism aims to reappropriate, on behalf of the workers and users, the latest developments in ICT applied to the hospitality and travel sector, in the form of, in this instance, collaborative online platforms. |
| **Territorial focus with global reach** | Although each offer is rooted in a specific territory, community and host, the content (offers, stories, content) can be freely exported to other digital media (digital badges, interoperability). |
| **Reinvestment of the essential surplus for general interest** | Profits are put into the creation of indivisible reserves and a research and development fund. |
| **Strategic vision** | The platform aims to meet a plural, inclusive demand for travel and accommodation experiences, in particular the trend towards coproduction with the traveller (preferred by 50% of users), and emerging purposes for travel: more fulfilling contact with locals, escape from standardisation, and so on. |

Source: elaborated by authors from [53,54] and [63–65].

It is clear that in the case of "Les Oiseaux de Passage", the principles involved in managing the tourism destinations are closely aligned with the criteria of good governance. This approach can also represent a means of tackling the complexity inherent in the governance of all tourist destinations, as illustrated above. Indeed, the model is scalable, and after little more than a year in operation, far from having stagnated, continues to grow and develop new destinations, whose communities have decided to join the cooperative (going from 7 to 40 destinations in just one year).

## 5. Discussion

*5.1. Territory as Commonly Held Heritage*

One of the crucial differences between the collaborative platforms developed under the auspices of the capitalist system and that of "Les Oiseaux de Passage" lies in the observation that the former are created and developed from outside the tourist destinations in response to purely economic interests, while the latter are driven from within the tourist locations themselves, such that their dynamics and their subsequent development respond to the stakeholders located in the destinations themselves, and seek to make these destinations areas of hospitality.

In fact, the cooperative belongs to the Faro Network, a collection of initiatives founded on the principles of the Council of Europe's Faro Convention. The network places emphasis on "heritage communities", and promotes collaborative and cooperative involvement of the hosts in finding creative ways of developing and managing this community heritage through the platform [66].

In this context, territory is considered a commonly held heritage asset, a common good [67], the result of how each civilization that has dwelt in the area has interpreted its relationship with the environment and its resources, and of how each generation reinterprets the signs and structures of its predecessors. From this perspective, the area is considered as living neo-ecosystem, in a constant process of territorialisation, de-territorialisation and re-territorialisation.

It is a vision which stretches far beyond the conventional economistic view of territory, on which the current concept of Smart Tourist Destination still rests, and which regards locations and spaces as mere technical supports of the economic system, thus reducing territory to a space for containing functions and circulations, an inert and isotropic support for economic activity. By contrast, in the vision of territory held by the cooperativists of "Les Oixeau de Passage", it ceases to be an abstract space of localisations of the masses and individuals, in which the cycles of life are independent of the identifying characters of the place which constitute its common heritage. Instead, it recovers these identities and opens them up to visiting travellers so that they can be participants in a living heritage in permanent transformation [54].

This enables this focus to rethink destinations not so much in terms of their potential visual beauty so much as their capacity to stimulate the rest of the senses. This is achieved by placing value on intangible heritage, by virtue of which the most marginalised places in towns and cities are often a rich resource (as is the case of the districts in the north of the city of Marseilles). Supporting tourist activity in territorial communities plays an important role in this regard, through the identification, appropriation and resignification of this heritage in the most marginalized social groups. Creativity, imagination and artistic endeavours represent a valuable resource in this process [68].

*5.2. Technology in the Service of Participative Relations between Humans*

Another key aspect of this collaborative platform, structured according to the principles and values of cooperativism and the social economy, is that its technological design is put to the service of the destination, the host communities and individuals, and the tourist experience, that is to say, the tourist.

The following advantages of this technological design can be highlighted for all those involved, promoting a more human contact and facilitating an alternative kind of tourist experience, one which is enriching both for the guest and their potential hosts [53,54]:

- Direct connection: the technological tool enables suppliers to promote the services and content they offer, authorising negotiation on a case by case basis. At the same time, by being based on annual subscriptions, it brings down the increasing costs of intermediaries (an average of 20% of the transaction on the large-scale platforms). This means that the costs of the platform are shared by and limited for the users.
- The technological tool also allows hosts and producers to present their offers in collaboration with their communities, as it is they themselves who best understand their location, while being able to

cooperate at the same time with other destinations, thus conferring on the platform a dimension of cosmopolitan localism [69].

- Any aspect which might harm human contact and foment competition on the platform is forbidden, including for example a points system, certificates, classification systems and so on.
- Every content item (offers, stories) can be freely exported to other digital media (digital badges, interoperability). This free broadcasting of data allows everybody to fully express their identity within a multitude of digital media: travel blogs, social networks, personal sites, online press, associated platforms and so on. It can hence be concluded that the management of the destination stretches from the most local—the communities—to the most global—the worldwide broadcasters on social networks, blogs, online platforms, and so on.
- The ergonomics favours personalisation and co-production of content, and provides fluent and poetic navigation around the website. Each member of the platform has their own page on which to present themselves. Narration is based on a collaboration tool and a simplified editor (texts and images) to give free rein to everybody's imagination while maintaining a common graphic identity. The travellers also have access to a simplified editor for sharing their impressions in visitor books and leaving testimony of their travel experiences.

In this regard, the predominating format of the major established collaborative platforms is deconstructed in a three-part process:

1. Disintermediation: facilitating direct interaction and interchange between hosts and visitors.
2. Deautomation: facilitating negotiation between both parties considering the social, environmental and economic contexts at the moment of exchange.
3. Destandardisation: facilitating the personalisation of experience and increased integration of the visitor in the local host space.

Consonant with this, the online reservation system is specifically designed to enable a "relational reservation system", something constituting not only an innovation in how prices are determined (no longer calculated by an algorithm balancing supply and demand, but directly as a negotiation between two humans), but also in how payment can be made, which allows for a wide variety of means (local currency, exemption from payment, bartering between companies, non-monetary exchanges, social aid and others).

In summary, under the guiding principles of cooperativism and the social economy, the platform is designed with the objectives of interaction, connectivity and the exchange of personal experiences between users always at the forefront. It does this in a horizontal and transparent manner, promoting creativity and enabling the expression of diversity in a shared space, encouraging mutual understanding and dialogue, and by doing so, enriching the experience of all those involved.

## 6. Conclusions

Analysis of the case-study presented in this paper makes a strong argument that the complexities of local governance in tourist areas can be successfully managed through the deployment of a methodological and axiological focus quite different to the paradigm that predominates in most of the literature consulted [46,60]. This alternative approach is founded fundamentally on the creation of genuinely cooperative networks and associations comprised of the stakeholders in the actual destinations themselves, and results in a more human and inclusive relationship with potential visitors.

The shift of focus from the current, consumption-oriented system to an innovative, hospitality-oriented focus, markedly more inclusive and human, requires a radically different understanding of how tourism might be configured, which breaks down the barriers of the United Nations classification and opens the service up to all kinds of travellers.

From a theoretical perspective, the case in question sheds light on the issue of sustainability in managing activities related to this new notion of hospitality within an area, and underlines the

importance of doing so at the point of supply. In other words, it is the stakeholders within the destination themselves who should coordinate and cooperate to provide travel, accommodation and activities, in combination with the opportunity to co-create an individualised experience on the part of the traveller or visitor.

This shift in focus also demands a new way of thinking about territory, one which considers it as a commonly held heritage asset, and which contributes to the reconstruction of places, interpreting the expressions of subjectivity that emanate from the new ways of inhabiting, welcoming and visiting through civic networks. It requires a renewal of how territorial heritage is interpreted and valued (in terms of the environment, landscape, human settlements, socio-cultural context) with respect to each location by its inhabitants in coproduction with the visitors.

The focus on cooperative governance is a tool with great potential for bringing together the different actors within an area to attempt to redress the dominance of hyperspace over the psychic and collective life of the location. Such actors include local residents, website operators, potential hosts, creative companies, freelance guides, cultural and environmental organisations, social initiative entities and any others interested in a shared interpretation of the territorial model, and in the well-being of travellers and residents beyond the merely economic.

The innovative research-in-action model, supported in part by medium-term public funding—particularly salient in this case—reflects the commitment of the state and justifies its claim to be an investor in innovation [70]. Moreover, it is worth noting that it enables fruitful dialogue between local expertise and professional insight. At the same time, it addresses the scalability (replicability) of the model, an aspect of crucial importance for an innovation developed on the margins of the system. This is achieved through facilitating an experimentation space to systematise learning, giving attention to the quality of the participatory model (scaling out), and giving importance to the cultural appropriation by participants in the system (scaling deep) [71].

Another significant factor in the question of replicability could also be the choice of model for digitization (scaling up) [71]. In this instance, dissemination is rendered relatively easy because the technology is simple. The platform facilitates direct interrelations, it is not an intermediary. This is important as it brings down the costs of maintaining the platform and managing reservations in comparison with other models such as that of Airbnb. Moreover, as it is open source, all communities involved can, if they wish, develop additional functionalities. It is a decentralised research strategy, open to innovation and promoting diversity of solutions while taking account of the plurality of local communities and resources. In addition to being favourable to the replicability of the model, it constitutes a source of permanent learning through collective intelligence.

The upshot of the foregoing is that many of the dehumanising effects often associated with such platforms are purged from the technology, in particular the lack of privacy and the loss of control over the nature of the host-guest relationship. In short, it is a question of reducing the involvement of artificial intelligence so as to increase the involvement of cooperative intelligence.

Another important consequence of this paradigm shift is that it can also evade the standardisation of the current model and its concomitant environmental costs. As illustrated by this study, this "seed" of a new model of tourism is a sound example of a SLOC (small, local, open and connected) Design [69], an innovative approach to travel and hospitality services and their associated territorial governance, whose scaling out (replication), scaling deep (cultural appropriation), and scaling up (diversity of stakeholders), if appropriately supported by public investment, could really transform the current model of tourism towards a more sustainable one.

The case study, then, illustrates a different way of understanding and doing tourism, one in which control of the development of the destinations, and with it control over the potential positive and negative impacts, is devolved to the local stakeholders.

The rules of governance—based on the cooperative social economy, and clearly recognisable and identifiable—, the use of open source, adaptable, low cost technology, and the clear willingness of the action research teams to disseminate the initiative that they are instrumental in promoting across

their extensive multi-actor, multi-level communication networks, are the characteristics that make the model easily imitated and hence replicable wherever a group of territorial agents share these values.

Nevertheless, against the theoretical coherence of the model, and the promising advances in its first year of operation, there needs to be set the limitations of this study. Chief among these is that the period it considers is too short to analyse the full range of implications concerning the advantages and disadvantages of this model of tourism governance, and in particular the issues regarding how the growth of the platform has been managed. This is especially so as this kind of cooperative governance is highly dependent on the context in which it is implemented and requires time for the changes in mentality in the relationships to be effected, and the trust networks between participants to be constructed.

In response to this limitation, the research teams have initiated a doctoral thesis project with the aim of identifying the complex relationships between territorial governance, tourism, the collaborative economy, heritage and cultural rights. This involves developing a methodology for investigating the impact of the model on the territories where it is implemented. By designing appropriate indicators, surveys and discussion groups (for collating data on the views of the members of the cooperative, the employment generated, the opinions of travellers, etc.) the project hopes to learn how to maximise the generativity, inclusivity and sustainability of the replication process.

This recently initiated line of research aims to address the principle limitation of the current study, regarding the short period of time since the commencement of the cooperative and the resultant lack of primary data sufficiently complete to allow an in-depth evaluation of its results.

**Author Contributions:** B.M.-U. and D.F.-R. conceived and designed the concept and outline for the article; they also reviewed the literature and wrote the manuscript; B.M.-U., D.F.-R. and P.W. analysed and described the case-study, and P.W. collected the data of the case analysis. P.W. supervised and made suggestions to the manuscript. All authors have read and agreed to the published version of the manuscript.

**Funding:** This research is the result of a transdisciplinary collaboration between two research projects: the PhD project funded by CIFRE—La convention industrielle de formation par la recherche of the Ministère de l'enseignement supérieur, de la recherche et de l'innovation (France) to the Centre d'Etudes en Sciences Sociales CESSMA, Université Paris Diderot,-Inalco,-IRD (201420722T) jointly with the Cooperative Les Oiseaux de Passage (Jan 2019-Dec 2021); and the Spanish R&D+I project "Educación Patrimonial para la Inteligencia Territorial y Emocional de la Ciudadanía". Ref: EDU2015-67953-P (MINECO/FEDER, UE).

**Conflicts of Interest:** The authors declare no conflict of interest.

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
