# Peer review of "Managing Tourist Destinations According to the Principles of the Social Economy: The Case of the Les Oiseaux de Passage Cooperative Platform"

_sustainability, doi:10.3390/su12124837_

Round 1

Reviewer 1 Report

The research work presented is of interest to the scientific community on tourism in the social economy. Furthermore, it has the potential to become a useful guide for practical application. The paper is well written and well structured. It is recommended that a number of minor modifications be made, which are described below:

- I believe that the objectives should be written in the introduction section.
- The authors comment on the methodology used before developing the theoretical framework. It is recommended that the methodology be included after the theoretical framework. Thus, it is recommended to remove the current section 2.
- It is recommended not to list the conclusions.
- In the conclusions section, it would be advisable to comment on the main practical contributions of the study, as well as to discuss the limitations of the study and future lines of research.

Author Response

Dear reviewer,

Firstly, the aim of the study has been included in the introduction (lines 77-90), while the methodology is presented in section 3, just before analysis of the case-study (section 4) and after the theoretical framework (section 2), that is lines 460-502.

Also in response to the suggestions the list of conclusions has been omitted and in its place we have added possible applications of the research, arguing that the case-study can be fully replicated in other destinations. Indeed, this is considered as one of the main contributions of the paper, and is developed in lines 808-813.

Reviewer 2 Report

Thank you for giving me the opportunity to review this paper, which I found very interesting and with a high potential for further development. Since I feel that this paper is well written, clearly addressed, and that it undertakes an interesting concern in the correct way, I would not add suggestions. Regarding this point, but ONLY if you might consider this, perhaps you could consider to include the figure for cruises on age 2 (line 54). Also, you would consider providing different scenarios for future research -according to different kinds of tourists or different localizations. And finally, I feel that in the introduction you place emphasis on the environmental side of sustainability. Nevertheless, I repeat that this is a very good job. Congratulations, and good luck!

Author Response

Dear reviewer,

With regard to the chief limitation of the study, as the lack of primary information, although the case-study presented in the paper is based in part on primary information (15 interviews with different stakeholders in the tourism sector by one of the authors, documented in two previous publications, both discussed in detail and fully cited in the paper) we accept that it has not been possible to develop a methodology specifically for collating the opinions of the members of the cooperative platform on its potential and limitations, chiefly because (among other reasons) it has been in operation for only a year and is still in its start-up stage in many of the destinations which have been incorporated.

Hence, at the suggestion of you we have added a paragraph to the conclusions indicating possible future directions for research: “… the research teams have initiated a doctoral thesis project with the aim of identifying the complex relationships between territorial governance, tourism, the collaborative economy, heritage and cultural rights. This involves developing a methodology for investigating the impact of the model on the territories where it is implemented. By designing appropriate indicators, surveys and discussion groups (for collating data on the views of the members of the cooperative, the employment generated, the opinions of travellers, etc.) the project hopes to learn how to maximise the generativity, inclusivity and sustainability of the replication process.” (lines 814-829).

Reviewer 3 Report

-The paper puts forward an ideological ground breaking model for sustainable tourism development with a powerful case study.

-It stresses the supply side of tourism however needed to be supported more strongly by primary and secondary research on how the demand side is changing to respond to this new model.

-Could have included interviews with the members of the cooperative  and the visitors and participants of the premises and activities.

Author Response

Dear reviewer,

With regard to the chief limitation of the study, identified by you as the lack of primary information, although the case-study presented in the paper is based in part on primary information (15 interviews with different stakeholders in the tourism sector by one of the authors, documented in two previous publications, both discussed in detail and fully cited in the paper) we accept that it has not been possible to develop a methodology specifically for collating the opinions of the members of the cooperative platform on its potential and limitations, chiefly because (among other reasons) it has been in operation for only a year and is still in its start-up stage in many of the destinations which have been incorporated.

Hence, at the suggestion of you we have added a paragraph to the conclusions indicating possible future directions for research: “… the research teams have initiated a doctoral thesis project with the aim of identifying the complex relationships between territorial governance, tourism, the collaborative economy, heritage and cultural rights. This involves developing a methodology for investigating the impact of the model on the territories where it is implemented. By designing appropriate indicators, surveys and discussion groups (for collating data on the views of the members of the cooperative, the employment generated, the opinions of travellers, etc.) the project hopes to learn how to maximise the generativity, inclusivity and sustainability of the replication process.” (lines 814-829)

Finally, as you note, although it can be considered appropriate, we chose not to explore demand according to primary and secondary sources, as to do so (at least so far as primary sources are concerned) would have meant a new research project with different aims and hypothesis. Nevertheless, the paper does make reference, albeit not exhaustive, to certain characteristics of demand in terms of bibliographic references which could be regarded as promoting models of tourism developed from the perspective of the offer, such as the fact that more than half of tourists make their holidays reservations online. These references are included in lines 102 and 114.